# Influence of Main Thoracic and Thoracic Kyphosis Morphology on Gait Characteristics in Adolescents with Idiopathic Scoliosis: Gait Analysis Using an Inertial Measurement Unit

**DOI:** 10.3390/s25144265

**Published:** 2025-07-09

**Authors:** Kento Takahashi, Yuta Tsubouchi, Tetsutaro Abe, Yuhi Takeo, Marino Iwakiri, Takashi Kataoka, Kohei Inoue, Noriaki Sako, Masashi Kataoka, Masashi Miyazaki, Nobuhiro Kaku

**Affiliations:** 1Department of Rehabilitation, Oita University Hospital, Oita 879-5593, Japan; kento0530@oita-u.ac.jp (K.T.); y-takeo@oita-u.ac.jp (Y.T.); marino58@oita-u.ac.jp (M.I.); t-kataoka@oita-u.ac.jp (T.K.); k-inoue@oita-u.ac.jp (K.I.); 2Division of Human Biology, Department of Health Science, Oita University of Nursing and Health Sciences, Oita 870-1201, Japan; 3Department of Orthopaedic Surgery, Faculty of Medicine, Oita University, Oita 879-5593, Japan; abe-te@oita-u.ac.jp (T.A.); noriaki-sako@oita-u.ac.jp (N.S.); masashim@oita-u.ac.jp (M.M.); nobuhiro@oita-u.ac.jp (N.K.); 4Physical Therapy Course of Study, Faculty of Welfare and Health Sciences, Oita University, Oita 870-1192, Japan; mkataoka@oita-u.ac.jp

**Keywords:** adolescent idiopathic scoliosis, spinal morphology, gait analysis, inertial measurement unit

## Abstract

**Highlights:**

**What are the main findings?**
In patients with adolescent idiopathic scoliosis (AIS), the main thoracic (MT) curve showed a positive correlation with mediolateral gait stability, whereas thoracic kyphosis (TK) showed a negative correlation.A gait symmetry analysis revealed distinct correlations between spinal curvature and mediolateral trunk acceleration across different quadrants.

**What is the implication of the main finding?**
Patients with AIS who have right-convex MT curves may experience a leftward shift in their center of gravity, thus affecting gait stability.A comprehensive understanding of the relationship between spinal deformities and gait patterns can inform the development of targeted rehabilitation strategies for patients with AIS.

**Abstract:**

This study examined the relationship between spinal morphological changes and gait characteristics in patients with adolescent idiopathic scoliosis (AIS) using inertial measurement unit (IMU) analysis. Twenty-three female patients with AIS scheduled for corrective surgery underwent a preoperative gait analysis using an IMU positioned at the third lumbar vertebra. Gait stability indicators were calculated, including root mean square (RMS) values for mediolateral (RMSx), anteroposterior, and vertical components. Peak mediolateral components in four coronal plane quadrants were also analyzed. Relationships with the main thoracic (MT) curve, the thoracolumbar (TL) curve, and thoracic kyphosis (TK) were assessed using Spearman’s rank correlation. The MT curve is positively correlated with RMSx, whereas TK exhibited a negative correlation. Gait symmetry analysis revealed a positive correlation between the MT curve and peak mediolateral trunk acceleration in the second (left upper) quadrant, and negative correlations for TK in the first (right upper) and fourth (right lower) quadrants. Patients with AIS who have right-convex MT curves demonstrated leftward center-of-gravity shifts, although reduced TK limited this compensatory mechanism. These findings may inform the development of rehabilitation strategies for AIS.

## 1. Introduction

Adolescent idiopathic scoliosis (AIS) is characterized by lateral spinal curvature developing during growth, predominantly affecting adolescents aged 12–18 years. AIS also involves lateral spinal deviation often accompanied by rotational deformities. The condition shows female predominance, and early diagnosis with timely intervention are crucial as progressive AIS can cause postural asymmetries, pain, and even compromise cardiopulmonary function [1]. While the etiology remains largely unclear, the contributing factors include genetic predisposition, neurodevelopmental abnormalities, and hormonal influences [2]. Asymmetrical bone growth and muscular imbalances during spinal development may contribute to scoliosis progression [3].

Conservative management with regular monitoring is typically recommended for mild scoliosis. For more severe cases, orthotic intervention and surgical correction are considered. Surgical treatment primarily involves spinal correction and fusion, with these procedures widely implemented [4]. Researchers have also increasingly investigated the efficacy of exercise-based interventions, such as the Schroth method and other conditioning approaches tailored to individual patient needs [5,6]. These therapeutic programs aim to enhance spinal flexibility, promote muscular balance, and improve trunk stability, effectively preventing scoliosis progression [7]. Early detection and intervention are particularly important for reducing the need for surgical intervention [8]. However, for exercise therapy to be effective, programs must be meticulously individualized based on a comprehensive assessment of spinal morphology and a thorough understanding of associated motor functions and movement patterns.

Center-of-gravity sway is reportedly increased in individuals with AIS compared to healthy controls, with a notable correlation between postural instability and thoracic curve progression [9,10]. While treatment strategies targeting gait pattern improvement in patients with AIS have been explored, the specific effects of physiotherapy, orthotic interventions, and surgical correction on gait mechanics remain controversial [11].

The inertial measurement unit (IMU) is increasingly used in clinical physiotherapy for quantitative and qualitative motor function assessment [12]. The IMU offers minimal environmental constraints, is user-friendly, and imposes negligible physical or psychological burden on patients. Consequently, it facilitates longitudinal assessments at regular intervals, providing valuable insights into qualitative changes in motor function over time. Despite this method’s limitations in analyzing specific gait phases, its low cost, ease of use, and straightforward data analysis enable its implementation regardless of institutional resources or specialized personnel availability. The importance of individualized rehabilitation has gained increasing recognition in recent years. Identifying indicators of latent gait characteristics requires the efficient accumulation of clinical data using simplified measurement techniques. Therefore, IMU-based gait analysis shows considerable promise for detecting gait characteristics in undiagnosed individuals with AIS within the community, enabling assessment beyond clinical settings. Furthermore, such data can provide a foundation for developing explainable artificial intelligence (AI) approaches using multivariate analysis and unsupervised learning to extract hidden gait features. Recent studies have demonstrated the use of data balancing techniques and generative AI for the scalable analysis of rare disease gait datasets, even with limited sample sizes [13].

Therefore, this study aimed to investigate the relationships between spinal morphological changes and gait characteristics in patients with AIS using IMU and to evaluate the clinical applicability of this method. Our findings may help expand treatment strategies and facilitate the development of tailored rehabilitation programs based on disease severity, ultimately improving patient outcomes.

## 2. Materials and Methods

### 2.1. Study Design and Participants

This single-center retrospective observational study was approved by the Ethics Committee of Oita University Faculty of Medicine (Approval No. 2889). The study included female patients with AIS aged 12–18 years hospitalized for corrective surgery at the Department of Orthopedic Surgery, Oita University Hospital, between August 2019 and May 2024. Exclusion criteria were (1) neurological symptoms from AIS affecting motor function or quality of life (QOL); (2) other orthopedic conditions potentially impacting motor function or QOL; and (3) history or current diagnosis of central nervous system disorders. As a result, 23 patients were included, with a mean age of 14.1 ± 1.14 years (Table 1).

### 2.2. Spinal Morphometry Using X-Ray Imaging

Standing X-ray images in the coronal and sagittal planes were used to classify curve patterns according to the Lenke classification system. The Lenke classification, defined by the Scoliosis Research Society, categorizes scoliosis into six curve types (Types 1–6) with lumbar modifiers (A, B, C) and sagittal thoracic modifiers (−, N, +) [14,15]. Three spinal morphology parameters were measured: the MT curve, the thoracolumbar (TL) curve, and TK. These parameters were determined by measuring angles formed by intersecting lines (Figure 1). The MT curve angle is formed by two tangential lines extending along the superior endplate of the uppermost vertebra and the inferior endplate of the lowermost vertebra within the thoracic region on the coronal plane [16]. The TL curve angle is similarly formed by two tangential lines extending along the superior and inferior endplates of the uppermost and lowermost vertebrae in the thoracolumbar region on the coronal plane [16]. As for TK, the angle is measured between two tangential lines extending along the superior endplate of the fifth thoracic vertebra and the inferior endplate of the 12th thoracic vertebra [17].

### 2.3. Gait Analysis Using IMU

An IMU (MVP-RF8-GC-2000, MicroStone, Kagawa, Japan) was positioned at the third lumbar spinous process and right calcaneal tuberosity. Participants walked barefoot during measurements. They walked at a self-selected comfortable speed along a 20 m walkway, including 5 m acceleration and deceleration phases. Data were recorded at a 200 Hz sampling frequency and transmitted in real time to a computer via Bluetooth. IMU functionality was visually verified. Following two trials, data from the second trial were analyzed. Trunk and right foot acceleration data were extracted across 10 gait cycles.

The following gait parameters were computed using Microsoft Excel: stride-to-stride time variability (STV) from right calcaneal tuberosity acceleration to assess gait cycle variability, root mean square (RMS) from trunk acceleration to evaluate gait stability, and Lissajous index (LI) to quantify gait symmetry [12,18,19,20]. RMS was computed separately for the mediolateral (RMSx), anteroposterior (RMSy), and vertical (RMSz) components. Overall gait instability was quantified using the composite RMS (RMSt). Since RMS is proportional to walking speed squared, RMS values were normalized by dividing them by walking speed squared, according to previous studies [12,21]. Additionally, LI was computed for the coronal (LIcor) and the transverse (LItra) planes. For LIcor computation, peak mediolateral acceleration values were extracted from four coronal plane quadrants (first: right upper; second: left upper; third: left lower; fourth: right lower) from the posterior view (Figure 2). Computation methods are as follows:

RMS(1)$$\mathrm{RMS}\;={\left(\frac{1}{T}\int_{t}^{t+T}a^{2}\left(t\right)\mathrm{dt}\right)}^{\frac{1}{2}}$$(2)$$RMSt=\sqrt{{RMS}_{x}^{2}+{RMS}_{y}^{2}+{RMS}_{z}^{2}}$$

*a*(*t*): acceleration signal; x: mediolateral direction; y: anteroposterior direction; z: vertical direction; *t*: total acceleration.

LI(3)$$LI=\left|\frac{2(R\;right-R\;left)}{\left(R\;right+R\;left\right)}\right|\times 100$$

R: rectangular area.

STV(4)$$STV=\frac{SD}{mean}\times 100$$

Mean: mean gait cycle duration; SD: standard deviation of gait cycle duration.

### 2.4. Statistical Analyses

Statistical analyses were performed using GraphPad Prism version 9.3. Data normality was assessed using the Shapiro–Wilk test, revealing non-normal distribution for some data. Therefore, correlations between spinal morphological parameters (MT curve, TL curve, TK) and IMU-derived gait characteristics were examined using Spearman’s rank correlation coefficient. Statistical significance was set at *p* < 0.05.

## 3. Results

### 3.1. Lenke Classification and Spinal Morphometry

Lenke classification distribution was as follows: Type 1, *n* = 11 (A−: 1, AN: 3, B−: 2, BN: 4, CN: 1); Type 2, *n* = 1 (BN: 1); Type 3, *n* = 2 (B−: 1, CN: 1); Type 4, *n* = 0; Type 5, *n* = 5 (C−: 1, CN: 4); Type 6, *n* = 4 (C−: 2, CN: 2) (Table 2).

The spinal morphometry revealed an MT curve of 39.1° ± 10.1°, a TL curve of 33.8° ± 10.7°, and a TK of 12.6° ± 9.2°.

### 3.2. Gait Analysis

The comfortable walking speed during IMU-based gait analysis was 1.21 ± 0.10 m/s (Table 2). The STV was 2.32% ± 1.14%. Gait stability parameters were as follows: RMSx 0.89 ± 0.16 m/s^2^, RMSy 1.54 ± 0.25 m/s^2^, RMSz 1.58 ± 0.25 m/s^2^, and RMSt 2.39 ± 0.27 m/s^2^. Gait symmetry indices for center-of-mass displacement were 16.1% ± 12.9% for LIcor and 17.8% ± 17.0% for LItra. Peak mediolateral accelerations in coronal plane quadrants were as follows: first 5.06 ± 2.25 m/s^2^, second 5.32 ± 1.98 m/s^2^, third 3.47 ± 1.70 m/s^2^, and fourth 3.36 ± 1.12 m/s^2^.

### 3.3. Correlation Between Spinal Morphometry and Gait Analysis

Table 3 presents the correlation coefficients and *p*-values between spinal morphometry and gait analysis parameters. The MT curve showed significant positive correlations with RMSx (*r* =0.536, *p* = 0.008) and the second quadrant peak (*r* = 0.463, *p* = 0.026) (Figure 3). TK showed significant negative correlations with RMSx (*r* = −0.550, *p* = 0.007), the first quadrant peak (*r* = −0.478, *p* = 0.021), and the fourth quadrant peak (*r* = −0.517, *p* = 0.012) (Figure 4). Conversely, the TL curve showed no significant correlations with any gait parameters.

## 4. Discussion

Most previous studies of gait characteristics in AIS primarily examined parameters such as speed and cadence, with limited exploration of their correlations with spinal morphological changes. This study is among the few kinematic analyses to use an IMU to investigate relationships between spinal morphology and gait characteristics in patients with AIS.

A gait instability analysis revealed an RMSx of 0.86 ± 0.16 m/s^2^, an RMSy of 1.54 ± 0.25 m/s^2^, and an RMSz of 1.58 ± 0.25 m/s^2^. Previous studies reported higher RMS values in healthy controls: RMSx at 1.51 ± 0.42 m/s^2^, RMSy at 1.96 ± 0.23 m/s^2^, and RMSz at 2.43 ± 0.47 m/s^2^ [22]. While direct comparisons are limited due to the higher mean age of the participants (mean age: 22.5 ± 3.8 years) in these studies, we initially hypothesized that individuals with AIS would exhibit higher RMS values than healthy controls. However, similar findings in the literature suggest that individuals with AIS may adopt a conservative balance control strategy [22,23]. Additionally, Mahaudens et al. proposed that this gait pattern is an energy-efficient strategy that reduces metabolic cost. Our findings support the hypothesis that individuals with AIS adopt a cautious, conservative balance strategy to maintain efficient and stable gait [24].

The observed relationship between gait instability and the MT reveals significant positive correlations between the MT curve and both RMSx and second quadrant peak values. Multiple factors, including aging and sensory impairments, contribute to increased lateral gait instability [25,26]. Additionally, impaired plantar sensation, as observed in stroke, increases lateral sway during walking [27]. Since this study examined young female subjects with AIS without neurological impairments, spinal morphological changes likely directly influence gait characteristics. AIS spinal deformities are characterized by coronal plane deviations, particularly in MT and TL curves. Consequently, the center of mass (CoM) in a static posture may be laterally displaced, which in turn influences CoM dynamics during movement. Notably, all participants had right-convex MT curves, suggesting rightward CoM deviation. During gait, pronounced MT curves may require greater compensatory lateral CoM shifts to maintain balance. The positive correlation between MT curve and RMSx likely reflects this compensatory mechanism. The significant positive correlation between the MT curve and the second quadrant peak supports this interpretation. Biomechanically, during left stance phase, the CoM must reposition within or near the left foot’s base of support for balance. With pronounced MT curves, rightward-displaced CoM requires amplified leftward shifts during this phase, increasing the second quadrant peak value. Patients with AIS reportedly show reduced stance phase duration, suggesting coronal plane CoM restrictions [28].

In contrast, the TL curve showed no significant correlation with gait instability. Meta-analyses and literature reviews on AIS gait characteristics indicate that coronal plane pelvic and hip movements during gait are reduced in AIS compared to healthy individuals [28,29]. Therefore, CoM displacement in AIS may be compensated for primarily by upper trunk movement (particularly the thoracic spine), rather than by the lower trunk (including the lumbar spine and pelvis). Thus, the TL curve may not directly influence gait instability.

TK, reflecting sagittal plane morphology, showed significant negative correlations with RMSx and the first and fourth quadrant peaks. The relationship between sagittal plane alignment and gait characteristics in AIS remains poorly explored compared to pelvic and hip alignment and their respective kinematics. Thus our findings elucidate the association between TK and gait dynamics, providing valuable insights into this underexplored area. Irrespective of AIS, kyphosis is the most common sagittal plane alignment abnormality. Kyphotic postural changes, often associated with aging and osteoporosis, are linked to deteriorating dynamic balance, increased postural sway, and gait instability [30,31]. Therefore, sagittal plane alignment may play a crucial role in gait disturbances. However, our study found an association between reduced TK and increased lateral sway, a finding potentially attributable to the reduced spinal mobility that accompanies TK reduction. While direct evidence linking TK reduction to decreased spinal mobility remains limited, prior research supports this association. For example, a study examining spinal morphology and respiratory function in adult patients with AIS identified MT curve flexibility—calculated as [(standing MT curve–lateral bending MT curve) ÷ standing MT curve × 100 (%)]—and TK as independent predictors of both percent-predicted forced vital capacity (%FVC) and forced expiratory volume in 1 s (%FEV1.0) [32]. Correlation coefficients between TK and %FVC (*r* = 0.512, *p* < 0.001) and between TK and %FEV1 (*r* = 0.509, *p* < 0.001) suggest that TK reduction contributes to reduced thoracic cage and spinal mobility. Given the relationships between the MT curve, RMSx, and the second quadrant peak, lateral CoM displacement in patients with AIS is counterbalanced by upper trunk movement. However, TK reduction with associated decreased spine and thoracic cage mobility may impair upper trunk compensatory mechanisms. Consequently, RMSx increases, raising the peaks in the first and fourth quadrants instead of the second, which is the typical direction for primary compensation. As this study provides no direct evidence to support this hypothesis, the interpretation remains speculative. Therefore, future studies incorporating respiratory function and thoracic spine mobility assessments are needed to further investigate this relationship.

In contrast, no significant associations were observed between LI and either the MT curve or TK. Our LI values were 16.1 ± 12.9% (coronal) and 17.8 ± 17.0% (transverse), lower than the median 23.0% (range: 1.2–55.4) reported in young adults (mean age: 22.1 ± 3.3 years) by Yamaguchi et al. [20]. However, high standard deviations in our LI values indicate considerable variability, consistent with previous research [20]. Therefore, LI may be somewhat unsuitable for capturing individual or disease-specific gait characteristics and did not reflect spinal morphological changes in this study. These findings suggest that the reliability and reproducibility of LI should be reevaluated before it can be used appropriately as an assessment tool in clinical or research settings.

This study has several limitations. First, the relatively small sample size may have reduced statistical power for detecting significant associations. The limited sample size resulted from the single-institution design and restricted number of surgical candidates. A priori power analysis using G*Power 3.1 estimated the required sample size for detecting a statistically significant correlation between two variables. Assuming a large effect size (ρ = 0.5), an alpha level of 0.05, and a statistical power of 0.80, analysis indicated a minimum requirement of 29 participants. Consequently, some findings may be subject to Type II error, limiting the generalizability of the results. Larger cohort studies are needed to validate and expand these findings. Second, the absence of age-matched healthy controls substantially limits the clinical interpretation of the findings. Although comparisons were drawn with healthy control datasets from previous studies in the discussion, the age mismatch between groups limits the validity of these comparisons and renders them insufficient as a basis for interpretation. Therefore, future studies should aim to compare AIS cases with age-matched healthy controls to more clearly identify gait characteristics specific to individuals with AIS. Third, participants were surgical candidates and thus may have exhibited relatively severe spinal morphological changes. In contrast, individuals with AIS qualifying for conservative management, such as exercise therapy, typically present with milder morphological changes in early disease stages. Future research should include patients with milder presentations not requiring surgical intervention. Additionally, our gait analysis derived an overall index for a single gait cycle without phase-specific analysis. For a more comprehensive assessment of CoM deviation effects, future studies should incorporate phase-specific analyses of the gait cycle.

## 5. Conclusions

As most patients with AIS have right-convex MT curves, our findings suggest that they achieve gait preservation through a compensatory mechanism: a pronounced leftward CoM shift to counteract an initial rightward displacement. A reduction in TK may limit these compensatory mechanisms by decreasing spinal mobility. While IMU-based gait analysis may not fully elucidate detailed gait biomechanics, it effectively captures gait characteristics in a simple, clinically applicable manner. These findings may contribute to the advancement of exercise guidance and rehabilitation protocols that are tailored to the MT and TK morphological features in individuals with AIS. Specifically, physical therapy targeting thoracic spine flexibility maintenance or improvement may improve gait stability.

Given the growing emphasis on explainable AI and personalized rehabilitation, future research should accumulate AIS gait data and incorporate multivariate analysis and unsupervised learning models to identify latent gait characteristics.

## Figures and Tables

**Figure 1 sensors-25-04265-f001:**
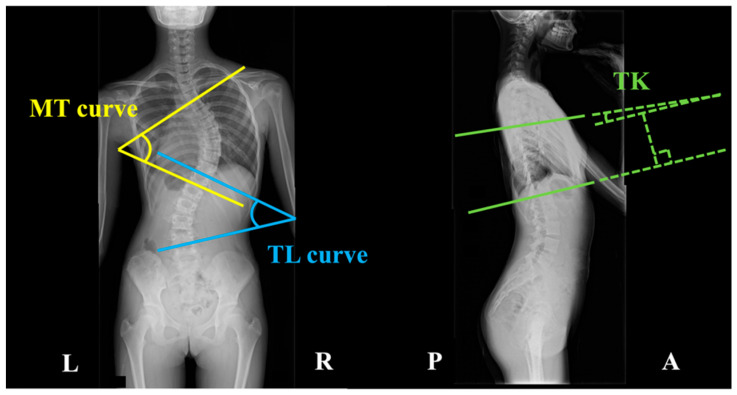
Measurement of spinal morphology with X-ray imaging. In a standing posterior–anterior radiograph, the angle is formed by two intersecting lines: the upper endplate of the main thoracic (MT) curve upper vertebra and the lower endplate of the MT curve lower vertebra. The thoracolumbar (TL) curve was also measured in the lumbar spine using the same method as the MT curve. In the sagittal plane, thoracic kyphosis (TK) spans T5 to T12. In a standing lateral X-ray image, the angle is formed by two perpendicular lines: the upper endplate of the T5 vertebra and the lower endplate of the T12 vertebra.

**Figure 2 sensors-25-04265-f002:**
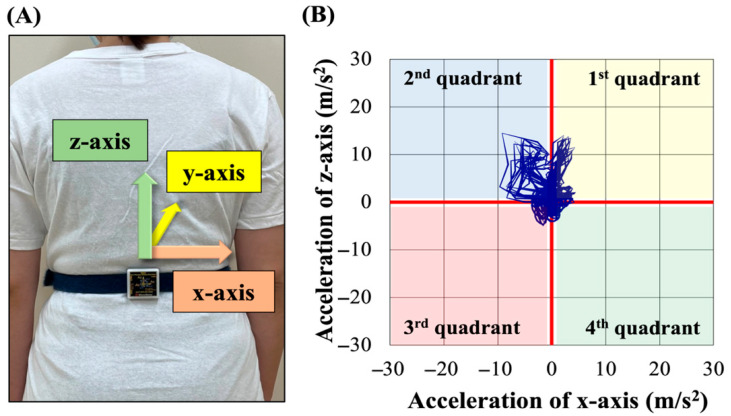
Attachment of the IMU and four quadrants of the frontal plane observed from the back. The *x*-axis represents the mediolateral direction; the *y*-axis represents the anteroposterior direction; and the *z*-axis represents the vertical direction (**A**). When observing the coronal plane from the back, it is delineated according to the Cartesian coordinate system (**B**). IMU: inertial measurement unit.

**Figure 3 sensors-25-04265-f003:**
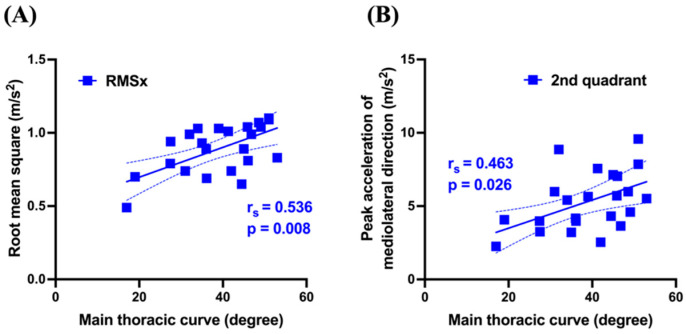
Correlation between the MT curve and RMS, as well as mediolateral peak acceleration in relation to the frontal plane. Solid line represents linear regression; dashed lines indicate the 95% confidence interval of the regression. Positive correlation between the MT curve and RMSx (**A**). Positive correlation with mediolateral peak acceleration in the second quadrant in relation to the frontal plane (**B**). MT: main thoracic curve; RMS: root mean square; x: mediolateral direction.

**Figure 4 sensors-25-04265-f004:**
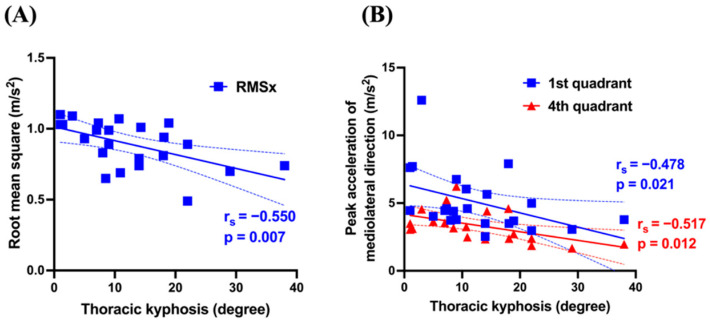
Correlation between TK and RMS, as well as mediolateral peak acceleration in relation to the frontal plane. Solid line represents linear regression; dashed lines indicate the 95% confidence interval of the regression. Negative correlation between TK and RMSx (**A**). Negative correlation with mediolateral peak acceleration in the first and fourth quadrants in relation to the frontal plane (**B**). TK: thoracic kyphosis; RMS: root mean square; x: mediolateral direction.

**Table 1 sensors-25-04265-t001:** Participant characteristics.

Characteristic	Variables
n	23
Sex (male/female ratio)	0:23
Age (years)	14.1 ± 1.14
Height (m)	1.54 ± 0.09
Weight (kg)	43.7 ± 6.38
BMI (kg/m^2^)	18.4 ± 1.72

BMI: body mass index. Mean ± standard deviation.

**Table 2 sensors-25-04265-t002:** Radiological parameters and IMU-based gait analysis results.

Lenke Classification	
Type 1 (A−:AN:B−:BN:CN)	11 (1:3:2:4:1)
Type 2 (BN)	1 (1)
Type 3 (B−:CN)	2 (1:1)
Type 4	0
Type 5 (C−:CN)	5 (1:4)
Type 6 (C−:CN)	4 (2:2)
Spinal morphological changes	
Main thoracic curve (degrees)	39.1 ± 10.1
Thoracolumbar curve (degrees)	33.8 ± 10.7
Thoracic kyphosis (degrees)	12.6 ± 9.2
Comfortable walking speed (m/s)	1.21 ± 0.10
Stride-to-stride time variability (%)	2.32 ± 1.14
Root mean square (RMS)	
x: Mediolateral direction (m/s^2^)	0.89 ± 0.16
y: Anteroposterior direction (m/s^2^)	1.54 ± 0.25
z: Vertical direction (m/s^2^)	1.58 ± 0.25
t: Magnitude of the RMS vector in three directions (m/s^2^)	2.39 ± 0.27
Lissajous index	
Coronal plane (%)	16.1 ± 12.9
Transverse plane (%)	17.8 ± 17.0
Peak mediolateral acceleration in the frontal plane ^#1^	
1st quadrant (m/s^2^)	5.06 ± 2.25
2nd quadrant (m/s^2^)	5.32 ± 1.98
3rd quadrant (m/s^2^)	3.47 ± 1.70
4th quadrant (m/s^2^)	3.36 ± 1.12

Mean ± standard deviation. ^#1^: The four quadrants are defined within the frontal plane as observed from behind. IMU: inertial measurement unit.

**Table 3 sensors-25-04265-t003:** Correlation coefficients between radiological parameters and IMU-based gait analysis.

	MT	TL	TK
Comfortable walking speed	0.300 (0.164)	−0.190 (0.386)	0.166 (0.451)
Stride-to-stride time variability	0.046 (0.835)	−0.023 (0.915)	−0.108 (0.625)
Root mean square (RMS)			
x: Mediolateral direction	0.536 (0.008)	−0.288 (0.183)	−0.550 (0.007)
y: Anteroposterior direction	0.349 (0.103)	−0.109 (0.622)	0.019 (0.930)
z: Vertical direction	−0.056 (0.800)	0.062 (0.778)	0.121 (0.582)
t: Magnitude of the RMS vector in the three directions	0.294 (0.173)	−0.134 (0.543)	−0.075 (0.733)
Lissajous index			
Coronal plane	0.223 (0.305)	−0.065 (0.769)	−0.167 (0.446)
Transverse plane	0.367 (0.085)	−0.052 (0.814)	−0.017 (0.939)
Peak mediolateral acceleration in the frontal plane ^#1^			
1st quadrant	0.377 (0.076)	−0.283 (0.190)	−0.478 (0.021)
2nd quadrant	0.463 (0.026)	−0.125 (0.570)	−0.187 (0.393)
3rd quadrant	0.387 (0.068)	−0.254 (0.243)	−0.288 (0.182)
4th quadrant	0.361 (0.090)	−0.048 (0.829)	−0.517 (0.012)

Spearman’s rank correlation was used to explore relationships. Figures in brackets indicate *p*-values. ^#1^: The four quadrants are divided by the frontal plane observed from the back. IMU: inertial measurement unit; MT: main thoracic curve; TL: thoracolumbar curve; TK: thoracic kyphosis.

## Data Availability

The data presented in this study are available on request from the corresponding author due to restrictions (e.g., privacy, legal, or ethical reasons).

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
