# Peer review of "Influence of Main Thoracic and Thoracic Kyphosis Morphology on Gait Characteristics in Adolescents with Idiopathic Scoliosis: Gait Analysis Using an Inertial Measurement Unit"

_sensors, 2025, doi:10.3390/s25144265_

Round 1
Reviewer 1 Report
Comments and Suggestions for Authors
This study investigated the relationship between spinal morphology (main thoracic curve MT, thoracolumbar curve TL, thoracic kyphosis TK) and gait characteristics in AIS patients through IMU gait analysis. The topic is clinically significant, and IMU technology aligns with current trends in biomechanical research. While the paper is well-structured, improvements are needed in the methods and interpretation of conclusions to enhance scientific rigor.
1、The title of the article highlights the overall impact of "spinal morphological changes", but the study actually focuses on the correlation between the main thoracic curve (MT), thoracolumbar curve (TL), thoracic kyphosis (TK) and gait parameters. The results show that only MT and TK are significantly associated with gait stability, while TL curve has no obvious correlation. It is suggested to adjust the title to "Influence of Main Thoracic and Thoracic Kyphosis Morphology on Gait Characteristics in Adolescents with Idiopathic Scoliosis", or clarify the specific reference to "spinal morphological changes" in the abstract and discussion to avoid disconnection between the coverage and the actual analysis.
2、Insufficient sample size and statistical power.: The study only included 23 female patients without providing a basis for sample size calculation. Given the heterogeneity of AIS patients (involving Lenke types 1-6), the current sample may not be sufficient to support subgroup analysis (such as correlation between types).
3、Lack of horizontal data comparison: The absence of a healthy control group limits the ability to determine whether the observed gait deviations are specific to AIS or represent general compensatory mechanisms in spinal deformities.
4、Non-normalized Data: Gait speed significantly influences acceleration metrics. While normalization to speed is mentioned (line 142), the actual normalization process should be explicitly detailed in results.
5、Insufficient guidance on rehabilitation: The conclusion section (lines 296-302) is vague. It is recommended to propose testable hypotheses (such as "Training for thoracic spine mobility may improve gait stability").
Reviewer 2 Report
Comments and Suggestions for Authors
- While IMUs are methodologically sound and appropriate for clinical gait analysis, the study is primarily descriptive and observational. The correlation-based design, which uses a small, homogeneous cohort (n=23, all female), limits the novelty. The authors should explicitly distinguish their work from previous studies that used optoelectronic systems or force platforms, emphasizing the novel insights provided by this IMU-based approach in addition to convenience and portability.
- The lack of a control group of age-matched healthy adolescents severely limits the findings' interpretability. While the primary goal may be to investigate within-group associations in AIS, it is unclear whether the reported gait asymmetries and RMS values are actually abnormal. Including, or at least referencing, normative datasets would help to contextualize the clinical relevance of the findings.
- The gait analysis appears to treat each walking trial as a single event, without breaking it down into discrete phases (e.g., stance, swing). This reduces biomechanical resolution, particularly since different phases are affected differently in AIS. The authors should either justify their choice or discuss how phase-specific segmentation could reveal more nuanced compensation patterns.
- Several speculative statements made in the discussion, such as the link between reduced thoracic kyphosis (TK) and decreased spinal mobility, are not clearly supported by the study's findings. These assertions should be explicitly presented as hypotheses or contextualized as directions for future research, which may include pulmonary function tests or direct spinal flexibility measurements.
- Although the Lissajous Index (LI) is discussed and calculated, its significance is minimized in both the results and the discussion. Given its significance in assessing trunk asymmetry in gait, the authors should clarify on how LI supports (or contradicts) the RMS and peak acceleration findings. As it stands, this metric is underleveraged.
- The caption for graphic 4 appears to be incorrect: it indicates correlations between the MT curve and RMSx, despite the fact that the graphic is about TK. This should be rectified. Furthermore, visual clarity could be increased by include confidence ranges or effect size annotations. Graphing significant p-values might also improve interpretability.
- Given the growing importance of explainable AI and tailored rehabilitation, the work could benefit from integrating (or suggesting) future modifications that use multivariate or unsupervised learning models to identify latent gait markers. For example, recent research has used data balancing and generative AI in rare illness gait datasets to provide scalable analysis even with tiny samples. The following studies may be used to support this path: Trabassi D, et al. Optimizing Rare Disease Gait Classification through Data Balancing and Generative AI: Insights from Hereditary Cerebellar Ataxia. Sensors 2024, 24(11), 3613. https://doi.org/10.3390/s24113613
This reference would underline the relevance of wearable-sensor-based gait analysis, especially in small, specialized groups, which is consistent with the current study's objectives.
Round 2
Reviewer 1 Report
Comments and Suggestions for Authors
The author has made revisions based on the previous review comments, addressing most of them. However, there is a lack of adequate response to the third review comment (Lack of horizontal data comparison: The absence of a healthy control group limits the ability to determine whether the observed gait deviations are specific to AIS or represent general compensatory mechanisms in spinal deformities). It is recommended that this issue be addressed before acceptance.
Reviewer 2 Report
Comments and Suggestions for Authors
I am satisfied with the responses, the work is now more robust and solid and deserves to be published.
Author Response
We sincerely thank you for your positive feedback and thoughtful evaluation.
We are pleased to know that the revisions have strengthened the manuscript and that you now consider the work to be more robust and solid.
Your constructive comments greatly contributed to improving the clarity and quality of our study.
Thank you again for your time and support.